# Identification and Source Attribution of Organic Compounds in Ultrafine Particles near Frankfurt International Airport

Florian Ungeheuer[1], Dominik van Pinxteren[2], Alexander L. Vogel[1,*]

[1]Institute for Atmospheric and Environmental Sciences, Goethe-University Frankfurt, Frankfurt am Main, 60438, Germany
[2]Atmospheric Chemistry Department (ACD), Leibniz Institute for Tropospheric Research (TROPOS), Leipzig, 04318, Germany

*Correspondence to:* Alexander L. Vogel (vogel@iau.uni-frankfurt.de)

**Abstract.** Analysing the composition of ambient ultrafine particles (UFP) is a challenging task due to the low mass and chemical complexity of small particles, yet it is a prerequisite for the identification of particle sources and the assessment of
potential health risks. Here, we show the molecular characterization of UFP, based on cascade impactor (Nano-MOUDI) samples that were collected at an air quality monitoring station nearby one of Europe`s largest airports in Frankfurt, Germany. At this station, particle-size-distribution measurements show enhanced number concentration of particles smaller than 50 nm during airport operating hours. We sampled the lower UFP fraction (0.010-0.018 µm; 0.018-0.032 µm; 0.032-0.056 µm) when the air masses arrived from the airport. We developed an optimized filter extraction procedure, used ultra-high performance
liquid chromatography (UHPLC) for compound separation, and a heated electrospray ionization (HESI) source with an Orbitrap high-resolution mass spectrometer (HRMS) as a detector for organic compounds. A non-target screening detected ~200 organic compounds in the UFP fraction with sample-to-blank ratios larger than five. We identified the largest signals as homologous series of pentaerythritol esters (PEE) and trimethylolpropane esters (TMPE), which are base stocks of aircraft lubrication oils. We unambiguously attribute the majority of detected compounds to jet engine lubrication oils by matching
retention times, high-resolution/accurate mass (HR/AM) measurements, and comparing MS/MS fragmentation patterns between both ambient samples and commercially available jet oils. For each UFP stage, we created molecular fingerprints to visualize the complex chemical composition of the organic fraction and their average carbon oxidation state. These graphs underline the presence of the homologous series of PEE and TMPE, and the appearance of jet oil additives (e.g. tricresyl phosphate (TCP)). Targeted screening on TCP confirmed the absence of the harmful tri-*ortho* isomer, while we identified a
thermal transformation product of TMPE-based lubrication oil (trimethylolpropane phosphate (TMP-P)). Even though a quantitative determination of the identified compounds is limited, the presented method enables the qualitative detection of molecular markers for jet engine lubricants in UFP and thus strongly improves the source apportionment of UFP near airports.

## 1. Introduction

Ultrafine particles (UFP) are particles with a diameter of less than 100 nm that are strongly influenced by primary emissions
in the urban environment (Allan et al., 2010). Current observations indicate that airports are major sources of UFP (Yu et al.,

2012; Keuken et al., 2015; Yu et al., 2017; Fushimi et al., 2019), but also road traffic and (biogenic) secondary organic aerosol are possible sources for nanoparticles contributing to the overall ultrafine particle mass in urban areas (Zhu et al., 2002a; Zhu et al., 2002b; Morawska et al., 2008; Paasonen et al., 2016, Rönkkö et al., 2017). Measurements in the vicinity of Schiphol airport (Netherlands) concluded that road traffic contributes less to the UFP burden compared to air traffic (Keuken et al.,
35  2015).

With increasing numbers of flight operations worldwide, aircraft emissions will become increasingly relevant regarding the long-term development of local and global air pollution and their impact on global radiative forcing (Lee et al., 2009; Masiol and Harrison, 2014; Yu et al., 2017). Forecasts indicate an increase of flights in Europe in the range of 1.9% per year with 16.2M movements in 2040 (Eurocontrol, 2018). Due to the corona pandemic however, the European flight traffic in 2020
declined by 55% compared to 2019 (Eurocontrol, 2021). Current forecasts predict a full recovery of flight movements between 2024 and 2029, depending on the pandemic course (Eurocontrol, 2020). UFP can be transported over long distances towards urban areas (Keuken et al., 2015; Hudda and Fruin, 2016), and although they have very low mass, they can cause oxidative stress, inflammatory reactions and other adverse health effects (Jonsdottir et al., 2019; Schraufnagel, 2020). In contrast to larger sized particles, UFP are able to penetrate the pulmonary alveoli, and can enter the bloodstream (Yang and Omaye, 2009).
At the same time studies indicate that UFP have an influence on humans via different mechanisms compared to $PM_{2.5}$, and are also able to create synergetic effects with transition metals (Costa and Dreher, 1997; Campen et al., 2001; World Health Organization, 2013). Still, the molecular composition of ambient UFP and their effects on human health is only poorly understood, primarily because of a lack of the deployment of specific and sensitive chemical measurement approaches.

Various studies examined the exhaust of aircrafts and found dependencies between particle emissions, their chemical
composition, engine design and the engine operating state. It is stated that the volatile organic fraction of the emitted particles is increasing through photochemical aging (Kılıç et al., 2018) and the oxidised organic fraction tends to condense on nucleation mode particles (Wey et al., 2007; Yu et al., 2017). The non-volatile fraction is dominated by black carbon. The ratio of volatile organics/non-volatile organics varies mainly depending on the operating state and the used jet fuel (Timko et al., 2010; Timko et al., 2013; Beyersdorf et al., 2014; Keuken et al., 2015). It appears that black carbon is a minor fraction of airport-related
particulate emissions (Keuken et al., 2015), by which it is often distinguished between the idle and the take-off status of aircrafts.

Further studies identified jet engine oil constituents in airport-located UFP (Yu et al., 2010; Timko et al., 2010; Yu et al., 2012; Fushimi et al., 2019; Yu et al., 2019). This is a consequence of the jet engine design, where rotating parts of the turbines need to be lubricated and other parts like bearings to be cooled. Due to a technically required venting system, nanometer-sized oil
droplets are released into the atmosphere (Yu et al., 2010). Depending on the seal tightness, the overall consumption of jet engine oil can amount up to 0.6 L/h/jet engine (Boyce, 2012), of which a part is emitted into the atmosphere.

The exposure of turbine and hydraulic oils to aircraft crews and ground-staff has been linked to health effects such as disorientation, headache, respiratory problems and weakness (van Netten and Leung, 2000; Winder and Balouet, 2001; Winder and Balouet, 2002). These symptoms, likely induced through an exposure with organophosphorous compounds are

summarised under the expression organophosphorus-induced delayed neuropathy (OPIDN). One possible etiological factor is the inhibition of the neurotoxic esterase (NTE) by organophosphorous compounds, which are present as additives in aircraft lubrication oil (Freudenthal et al., 1993; Ehrich et al., 1993; Eyer, 1995; O'Callaghan, 2003). However, the understanding of health effects from potentially harmful chemicals formed during aircraft operations requires further elucidation, as it has been shown that e.g. tricresyl phosphate exposure from bleed air in aircraft cabins is below effect thresholds and cannot explain

observed symptoms (Boer et al., 2015).

Therefore, it is important to determine the chemical composition of UFP near airports in order to identify their sources during airport operations, to assess further possible detrimental effects on human health, and finally, to mitigate their emission. Due to the low mass of UFP, the typical sampling and extraction methods need to be refined in order to characterize particles with aerodynamic diameters < 56 nm, requiring detection limits of low picogram levels of single organic molecules. Here, we

describe an offline method combining size-resolved UFP sampling with target and non-target screening by ultra-high performance liquid chromatography (UHPLC) coupled to high-resolution mass spectrometry (HRMS) in order to shed light onto the chemical composition of airport-related UFP. The overall aim of this study was to develop an analytical routine that allows identifying as many as possible compounds contributing to the particle mass. Within the scope of this study, we compare the molecular composition pattern of 22 size-resolved ambient UFP samples with the composition of five different jet engine

lubrication oils. Therefore, our results provide an overview about the chemical composition of UFP near large international airports, covering various aircraft engine designs, operating conditions, fuels and jet engine lubrication oils.

## 2. Methods

### 2.1 Frankfurt airport – Measuring site

Frankfurt airport is one of the largest airports in Europe with more than 500,000 flight operations in 20182019, shared over

four runways. It is located in the Rhine-Main metropolitan area within a distance of around 12 km to the city centre of Frankfurt. In 20182019 more than 6970.5M passengers and 2.3M1M tons of cargo have been transported with a consumption of around $5.5 \times 10^6$ m$^3$ of kerosine (Fraport AG, 2020). The corona pandemic caused a decline in flight movements by 58.7% in 2020 compared to 2019. The transported cargo decreased by 8.5% and passenger numbers by 73.4% (Fraport AG, 2021).

Since September 2017 the size distributions of ambient particles, including the ultrafine fraction, are monitored at an air

quality monitoring station of the Hessian Agency for Nature Conservation, Environment and Geology 4 km north of Frankfurt airport (Figure 1). It is located in a forested area with no highly frequented streets within 1 km distance. This sampling station is considered representative for the outflow emissions of highly-frequented airports.

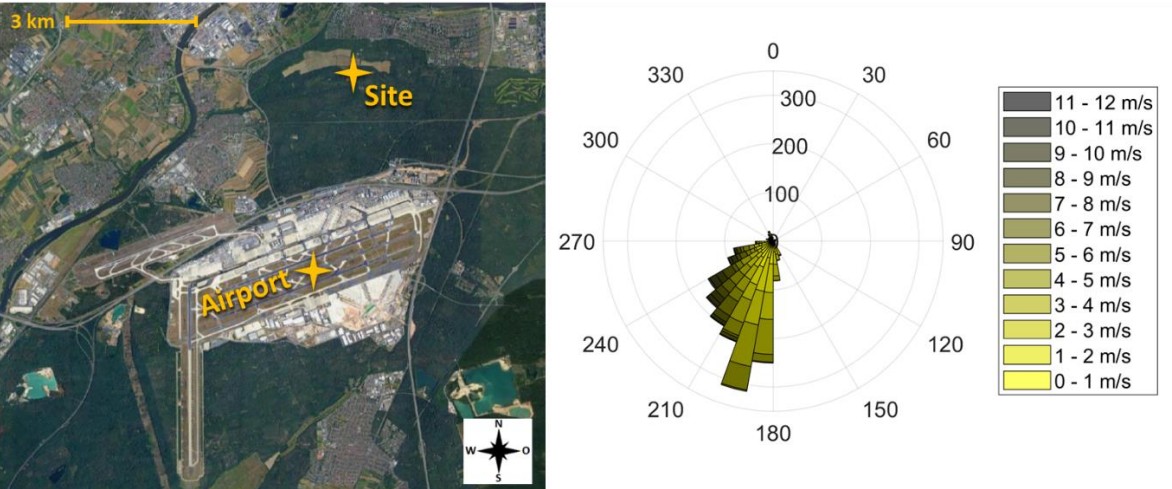

Figure 1. Illustration of the sampling site in a distance of 4 km to Frankfurt airport (Pictures © 2020 AeroWest, GeoBasis-DE/BKG, GeoContent, Landsat/Copernicus, Maxar Technologies, Mapdata © 2020 Google). The wind-rose indicates the wind direction during the eight sampling periods of the campaign. Data provided by the meteorological station at Frankfurt airport (ICAO-Code: EDDF) of the Deutscher Wetterdienst (DWD).

## 2.2 Sampling

We used a Micro Orifice Uniform Deposition Impactor (Nano-MOUDI, Model 115, MSP, Minneapolis, MN, USA) which consists of a 13 stage impactor system that is able to collect size-resolved UFP on the last four stages. However, we only sampled on the last three stages (aerodynamic diameters between 0.010-0.018 µm, 0.018-0.032 µm and 0.032-0.056 µm), since particle size distribution measurements show the highest number concentrations for particles smaller than 50 nm (Rose and Jacobi, 2018). This type of low pressure cascade impactor was already used by other research groups for analysis of atmospheric and industrial particulate matter (Cena et al., 2014; Chen et al., 2016; Fushimi et al., 2019).

The Nano-MOUDI was installed at the air-quality monitoring site in Frankfurt Schwanheim (Figure 1). We sampled ambient air without a pre-impactor inlet through a stainless steel tube (ID 1 inch, length 2 m), towards the Nano-MOUDI with a total sampling flow rate of 1.8 $m^3$/h. The sampling on the last three stages was conducted with a flow rate of 0.6 $m^3$/h. The deposition of particles >0.056 µm on the upper ten stages and the sampling of ambient UFP on the last three stages was carried out using aluminum foils (TSI, Ø×THK: 47×0.015 mm). To limit a bounce-off of larger particles to the lower stages, the foils for the upper ten stages were coated with Apiezon® grease L. Therefore, after baking out the filters for 30 min at 300°C, a 1 cm section of the Apiezon® grease was dissolved in 100 mL *n*-hexane, and each foil was evenly coated by adding six droplets of the mixture. After drying for 24 hours in a fume hood, each single foil was stored in a glass petri dish. The foils for the UFP sampling were heated at 300 °C for 30 min and then stored in metal boxes until sampling.

Sampling took place from August to October 2019 exclusively during southerly wind direction and during the airport operating hours (5:00 – 23:00 CET). Sampling times differed between 18 and 54 hours (Table S1) in order to accumulate sufficient mass for analysis. Shortly after sampling, the filters were stored in metal boxes at −20 °C to prevent subsequent evaporation or

transformation processes. The final filter loadings depend on sampling time, meteorological conditions and operating state of the airport. Although the particle emissions of aircraft engines depend on the actual thrust (Timko et al., 2014; Lobo et al.,

2015; Yu et al., 2017), we assume that we sampled a mixture of UFP that represent different aircraft engines under various operating states. Hence, the collected samples reveal the average UFP emissions of the whole airport and not of individual engines or operating conditions.

We collected field blanks on the last three stages for 115 hours without active sampling air flow, and processed them analogously to the UFP samples.

**2.3 Filter extraction**

In consequence of the low mass of the sampled UFP (sub µg range of total particle mass after 30-50 hours of sampling), the extraction method was developed in order to achieve the highest possible extract concentration. Simultaneously, no pre-concentration step through solvent evaporation was conducted in order to prevent volatilization of target compounds. Owing to the strongly adsorptive behaviour of the jet engine oil constituents, no filtration step of the solvent extracts was implemented.

Various solvents were tested regarding their ability to dissolve the jet engine oil target compounds and the overall particle mass. Pure organic solvents provided the higher extraction efficiency than mixtures with water and similar ~~ones to mixtures of acetonitrile and methanol.~~efficiencies to mixtures of acetonitrile and methanol. The tests were based on UFP filter extractions with different solvents (100% methanol; 50% acetonitrile 50% methanol; 60% methanol 40% water; 60% acetonitrile 40% water), and evaluated based on the non-target-analysis-generated identifications and corresponding signal

intensities. Due to the use of methanol (Optima™ LC/MS grade, Fisher scientific) and water (Milli-Q, Merck) as the UHPLC solvents, pure methanol was used for the filter extraction.

A circular section with a diameter of 2.5 cm was cut out of each foil sample ~~according to~~located below the array of the nozzles of each impactor stage. It is worth mentioning that the region of deposited UFP on the foil surface was generally not visible. The resulting sections were cut in small pieces (approximately 2.5mm x 2.5mm) with a ceramic scissor and extracted two

times with 100 and 50 µl, respectively, for 20 minutes each on an orbital shaker (300 rpm). We avoided ultrasonic extraction due to a possible formation of free radicals, which can lead to chemical transformation of the original sample composition (Riesz et al., 1985). The extraction was conducted in glass vials equipped with flat bottom micro inserts with a maximal volume of 200 µl (LLG Labware, Ø×H: 6×31 mm), to ensure a complete filter surface covering, and then pipetted into 100 µl micro inserts with conical bottoms (VWR, Ø×H: 6×31 mm), to guarantee a complete UHPLC injection of 5 µL in the autosampler.

Altogether 22 filter samples and three field blanks were analysed.

**2.4 Jet engine lubrication oils**

Jet engine oils are designed of a base stock material like pentaerythritol esters or ~~trimethyolpropane~~trimethylolpropane esters and different additives. Organophosphate additives are used as anti-wear agents and metal deactivators (Wyman et al., 1987; Du Han and Masuko, 1998), whereas amine constituents serve as stabilizers (Wu et al., 2013) (Table S2).

In order to characterize UFP and screen for the molecular composition of jet engine oil, we analysed five different jet engine oils of various brands. The oils were selected regarding the recommended oil brand of engine manufacturers. The Mobil Jet Oil II (ExxonMobil, Irving, TX, USA) has the highest market share (Winder and Balouet, 2002), whereas the market share of the other oils (Mobil Jet Oil 254 (ExxonMobil, Irving, TX, USA), Aeroshell 500 (Royal Dutch Shell, The Hague, Netherlands), Turbo Oil 2197 and 2380 (Eastman, Tennessee, USA)) is unknown.

## 2.5 UHPLC/HRMS method


The chemical characterization of the airport filter samples was realized by using ultra-high performance liquid chromatography (UHPLC) (Vanquish Flex, Thermo Fisher Scientific)/heated electrospray ionization (HESI) hyphenated with an Orbitrap high-resolution mass spectrometer (HRMS) (Q Exactive Focus Hybrid-Quadrupol-Orbitrap, Thermo Scientific) as a detector. The chromatographic separation was achieved by using a reversed phase column (Accucore $C_{18}$, 150 x 2.1 mm, 2.6 µm particle

size, Thermo Fisher Scientific), operated in gradient mode, and thermostated at 40 °C (still air). Ultrapure water (18.2 MΩ·cm, Millipak® Express 40: 0.22 µm, Millipore; Milli-Q® Reference A+, Merck) with 0.1% formic acid (v/v, solvent A) and methanol (Optima™ LC/MS grade, Fisher scientific) with 0.1% formic acid (v/v, solvent B) were used as the UHPLC solvents. Formic acid (LiChropur®, Merck) with a purity of 98-100% was used to improve the chromatographic separation and ionization in the positive mode. Due to the high fraction of nonpolar compounds in the samples, the UHPLC chromatography

started with 60% solvent B (0-0.5 min), was then increased to 90% (0.5-11 min) and then ramped to 99% (11-16 min). The high-organic-solvent starting conditions allow the injection of large sample volumes of organic solvents, without compromising the chromatographic separation. In the end, solvent B was reduced to 60% (16-17 min) and the system was allowed to equilibrate for the subsequent measurement within three minutes. The overall method duration was 20 minutes with a flow rate of 400 µl/min and an injection volume of 5 µl. The ionization for the mass spectrometric detection was conducted

by HESI in positive and negative mode. The HESI settings in positive mode were: 3.5 kV spray voltage, 40 psi sheath gas (nitrogen), 8 psi auxiliary gas (nitrogen) and 350 °C gas temperature. The negative mode was operated with the same settings but with a spray voltage of 2.5 kV. The scan range in the positive mode was from 150 to 750 and in the negative mode from 50 to 700 mass-to-charge ratio (*m/z*) with a resolution of ~70k at *m/z* 200. In negative mode molecular ions ([M-H]$^{-}$) are produced by deprotonation, whereas in positive mode molecular ions ([M+H]$^{+}$; [M+Na]$^{+}$) are generated by adduct formation.

In order to gain structural information of the single molecules, a data-dependent MS/MS (dd-MS$^{2}$) method was used with MS-data recorded in profile mode.

The limit of detection (LOD) and limit of quantification (LOQ) of tri-*o*-cresyl phosphate (TCP) and pentaerythritol tetrahexanoate ($C_{29}H_{52}O_8$) was determined by external calibration according to the norm 32645 of the German Institute for Standardization (DIN). For tri-*o*-cresyl phosphate (≥97.0%, Sigma Aldrich) and for pentaerythritol tetrahexanoate (95%,

Carbosynth Ltd) we calibrated with five points in the range of 0.001 – 1 ng/µL. Each calibration point was measured three times in succession. We determined the LOD of pentaerythritol tetrahexanoate as 0.007 ng/µL and the LOQ as 0.018 ng/µL. For tricresyl phosphate we achieved a LOD of 0.021 ng/µL and a LOQ of 0.060 ng/µL. Both calibration curves show only a

small linear response range, which is likely due to their adsorptive behaviour on glass surfaces. Therefore, we used only three concentrations in the range of 0.001 – 0.1 ng/µL for the determination of the LOD and LOQ. Presumably, all the detected

pentaerythritol esters and trimethylolpropane esters show a similar adsorption tendency like the used ester standard. Due to the adsorptive behaviour, quantification of the UFP constituents in ambient samples should preferentially be carried out by standard addition. We translate the results of the calibration into ambient concentration LODs: an average sampling duration of 40 h with the Nano-MOUDI flow rate of 0.6 m$^3$/h results in a sampled air volume of 24 m$^3$. Reaching the LOD of pentaerythritol tetrahexanoate and tricresyl phosphate in a solvent extraction volume of 150 µL would require ambient air

concentrations of 44 pg/m$^3$ and 131 pg/m$^3$ (for each sampled size interval), respectively. However, we can project this calculation on ambient samples only under the assumption that no losses occur during sampling.

## 2.6 Non-target screening

The full scan MS-spectra analysis was carried out by the non-target analysis software Compound Discoverer (CD) (version 3.2.0.261, Thermo Fisher Scientific), which was already validated regarding the qualitative and quantitative results by

comparison with the non-target analysis software MZmine2 (Vogel et al., 2019). The compound discoverer software identifies substances and determines their molecular formula depending on their exact mass, isotopic signature, MS/MS fragmentation pattern and by database matching of MS/MS fragmentation spectra (database: www.mzcloud.org).

We iteratively optimized the non-target analysis workflow (see Figure S1 and Table S3 for the detailed settings of the CD-workflow) in order to achieve the best possible characterization of the particles chemical composition. The described jet engine

oils were diluted to a concentration of 1 µg/mL in methanol (Optima$^{TM}$ LC/MS grade, Fisher scientific) and analysed with the identical non-target approach. The chromatograms of the jet engine oils, the UFP samples and the field blanks were processed in one experiment in order to improve the software-based identification procedure. We filtered the non-target analysis results with the following criteria: the sample signal must be more than five times larger than the blank signal, the minimum area is 1.0 x 10$^5$, and the minimum retention time is 0.7 min (column dead time peak at 0.6 min). A large signal at m/z 399.2506 ± 4

ppm ([M+H]$^+$ of Tris(2-butoxyethyl) phosphate, C$_{18}$H$_{39}$O$_7$P) was manually filtered out prior to the graphical illustration because of its ubiquitous use as plasticizer, flame retardant and floor finish/wax (Lewis, Richard J., Sr. et al., 2016). The peak areas of the compounds in the samples were background corrected by subtracting the corresponding peak areas of the field blank. For the detection of the compounds, a mass tolerance of 5 ppm was allowed, whereas we reduced the mass tolerance for the molecular formula prediction to 2 ppm. The allowed elemental combinations for the peak identification were defined

as C$_1$ H$_1$ to C$_{90}$ H$_{190}$ Br$_3$ Cl$_4$ N$_4$ O$_{20}$ P$_1$ S$_3$. Finally, we classified the assigned molecular formulas into composition groups (CHO, CHN, CHNO, CHOS, CHNOS, CHOP, other), which help to graphically illustrate the complex chemical composition of UFP.

## 2.7 Fragmentation experiments

In order to gain structural information of the chemical species found by the non-target analysis, we conducted successive data-dependent full-scan/tandem-MS (dd-MS$^2$) measurements in the discovery and confirmation mode, respectively. In a first measurement, we used the discovery mode to obtain the exact mass and possibly MS/MS spectra of abundant signals. In this mode, the mass spectrometer estimates a peak apex during acquisition of the chromatogram, and triggers the recording of a fragmentation spectrum by higher-energy collisional dissociation (HCD) of the high abundant signals. Subsequently, we analysed the discovery-mode data with CD, and we obtained a list of compounds that are significantly different from the blank. This list was loaded as an inclusion list for a second measurement of the samples. By this approach, we recorded high-quality MS/MS fragmentation spectra of the listed compounds-of-interest (even of the low intensity compounds), allowing us to confirm the molecular structure and improve the accuracy of the database identification process. The MS/MS experiments were conducted with a normalized collision energy (NCE) of 30.

We used the different jet engine lubrication oils as qualitative identification standards, with regard to the material safety data sheets (MSDS) declaring the composition of the oils. With help of the exact mass, retention time and MS/MS fragmentation pattern of the jet oil constituents, we were able to unambiguously identify the oil-related compounds in ambient UFP samples. In the case of TCP, the different structural isomers do not show different MS/MS fragmentation pattern. Here, we make use of different retention times for the unambiguous identification of tri-*o*-cresyl phosphate, enabling us to examine whether the *ortho*-isomer can be found in the jet oils and the ambient UFP samples.

## 3. Results

### 3.1 Non-target screening and molecular fingerprints of UFP

The non-target screening detected almost 1000 compounds in the positive ionization mode. However, the majority of these compounds does not distinguish from the blank. Still, we find approximately 200 organic compounds at a sample-to-blank ratio >5 in the largest size fraction (0.032 – 0.056 µm), averaged over all 22 filters of this size fraction. The two smaller size fractions yielded more than 100 organic compounds with a sample-to-blank ratio >5 (Figure 2). Averaged over all filters of the largest size fraction, we obtained sample-to-blank ratios >100 for approximately 30 compounds in the positive ionization mode. The particles in the size fraction 0.032-0.056 µm show a five to ten times higher signal intensity of the most abundant compounds, as well as a larger number of compounds, due to the larger mass concentration compared to the smaller size bins. The negative ionization mode revealed only 16 compounds with the same data processing filter criteria.

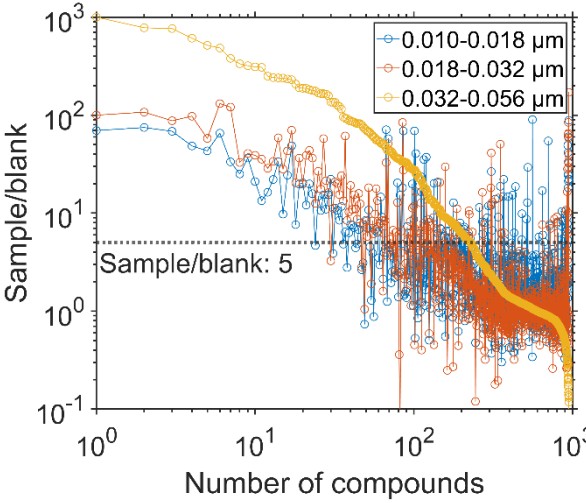


**Figure 2. The sample-to-blank ratios of the detected compounds, averaged over all samples belonging to a certain size fraction (0.010-0.018 µm; 0.018-0.032 µm; 0.032-0.056 µm).** **Detection of the compounds was accomplished in positive ionisation mode.** **The sample-to-blank ratio of 5 was specified for data filtering and is displayed as a dashed horizontal line.**

For each size fraction, we use the molecular fingerprints (retention time vs. molecular weight (MW), Van-Krevelen-diagram,

Kroll-diagram, and Kendrick mass defect vs. MW) for illustration of the chemical composition of the organic fraction, which

is shown in Figure 3 for 0.032-0.056 µm particles. Molecular fingerprints of the size stages 0.010-0.018 µm and 0.018-0.032

µm are shown in Figure S2 and Figure S3, respectively. Each circle represents a compound, and the colouring describes the

molecular composition group. Signals to which a molecular formula could not be assigned are classified as "other". The area

of the circles is proportional to the measured signal intensity. The measured mass-to-charge ratios (*m/z*) are converted into

MW.

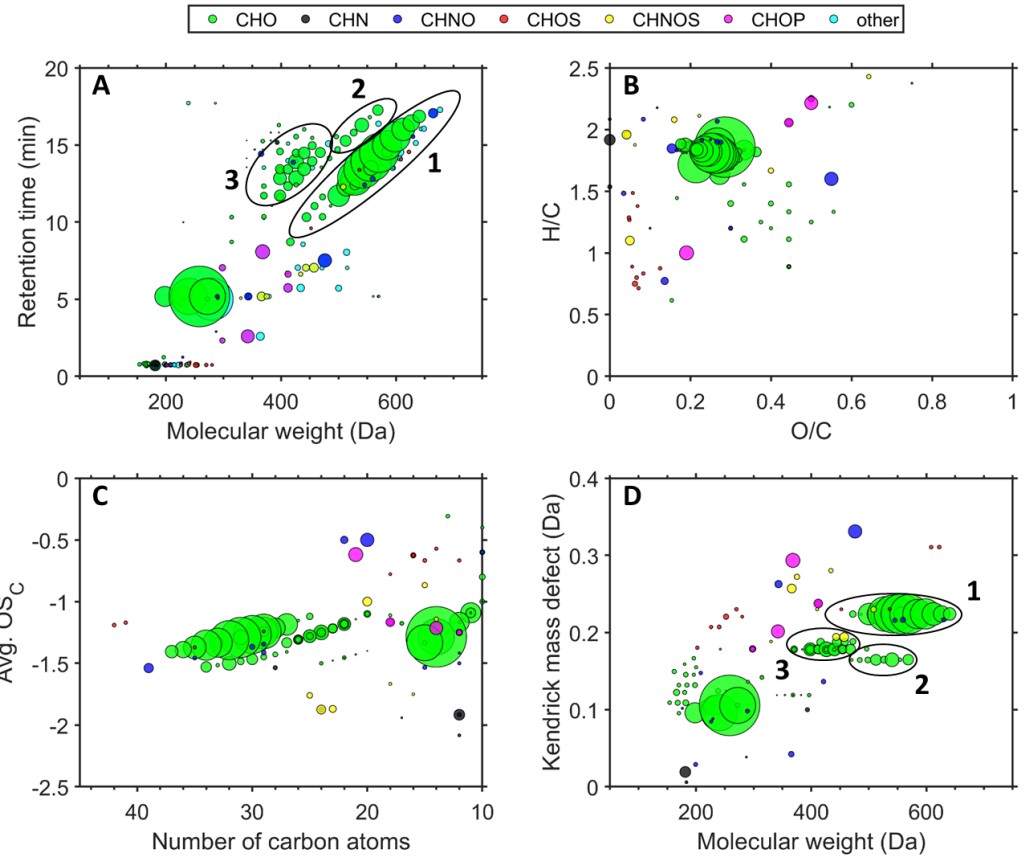

**Figure 3. Molecular fingerprints (Retention time vs. MW [A], Van-Krevelen-diagram [B], Kroll-diagram [C], Kendrick mass defect vs. MW [D]) of six averaged airport-related ultrafine particle samples in the size range of 0.032-0.056 µm.**

Figure 3A (retention time vs. MW) shows primarily CHO-compounds. Group 1 represents mainly the homologous series of

pentaerythritol esters ($C_{27-38}H_{48-70}O_8$). We found that Compound Discoverer falsely interprets ion signals as $[M+H]^+$ of the measured ions $[M+Na]^+$, $[M+K]^+$ and $[M+NH_4]^+$ of the pentaerythritol esters, and hence suggests up to three likely false molecular formulas per ester molecule. These artifacts were manually removed, based on the same retention time and exact mass difference to the molecular ions of the pentaerythritol esters. The native molecular fingerprint is displayed in Figure S4. Pentaerythritol esters are used as a common base stock material for synthetic jet engine lubrication oils (Winder and Balouet,

2002), and therefore can be attributed to jet engine oil emissions. These synthetic oils are particularly used in aviation, while the base stock of automotive oils is made of a crude oil fraction that consists mostly of petroleum hydrocarbons (Vazquez-Duhalt, 1989). Another jet engine oil base stock material of trimethylolpropane esters ($C_{27-34}H_{50-64}O_6$) (Wright, 1996) forms a homologous series in group 2. Group 3 can be mainly attributed to fragmentation, probably formed during the ionization of the pentaerythritol esters in the HESI source. The compounds in group 3 have the same retention times as the original

pentaerythritol esters of group 1, but differ in their molecular weight, which can be explained by the neutral loss of fatty acid

fragments (e.g. $C_5H_{10}O_2$, $C_7H_{14}O_2$, $C_8H_{16}O_2$ and $C_{10}H_{20}O_2$). Although electrospray ionization is known as a very soft ionization technique, we observe different fragmentation products of the pentaerythritol esters, depending on the side chain length. In order to minimize fragmentation of the pentaerythritol esters, we tested different settings of the HESI source: The auxiliary gas heater temperature was reduced from 350 °C to 200 °C in 50 °C steps and the spray voltage was reduced from 3.5 kV to

2.0 kV in 0.5 kV steps. Variation of both parameters, which were identified as the most important driver of the ionization process, did not reduce fragmentation of the pentaerythritol esters. Besides the large CHO-compounds at 5.2 min, which can likely be attributed to decanedioic acid-1,10-diethyl ester ($C_{14}H_{26}O_4$) and its fragments, we do not detect large contributions of other organic species in this size fraction. Only a small number of signals remains unidentified, appearing in the "other" composition group. Hence, the results of the non-target analysis suggest that the sampled UFP are mainly composed of

lubrication oils. However, other techniques might reveal the presence of additional compounds, (e.g. metals, black carbon, inorganic compounds, etc.), which are not detected by the presented technique.

The Van-Krevelen-diagram describes the different compounds regarding their hydrogen to carbon (H/C) and oxygen to carbon (O/C) ratios (Figure 3B). Compounds that consist mainly of single-ring aromatic moieties have an H/C ratio around one. Tricresyl phosphate, an anti-wear additive in synthetic jet oil, contains three single-ring aromatic (cresol) moieties. TCP

appears in Figure 3B as a magenta circle at H/C = 1 and O/C = 0.2. The majority of the detected compounds however, are located in the upper part of the diagram (H/C > 1.5), which implies that most of the airport-related UFP do not have an aromatic character. With O/C ratios below 0.6 it can also be stated that the UFP do not become oxidized during the 4 km transport distance between the emission source and the monitoring station. The analysed jet engine lubrication oils show O/C ratios within the same range.

The average carbon oxidation state (avg. $OS_C$) versus the number of carbon atoms (Figure 3C) is a way of characterising the oxidation state of complex organic compound compositions, and is calculated according to Kroll et al. (2011). The diagram shows that the synthetic esters (CHO) are large molecules containing 27-37 carbon atoms, which implies a low vapour pressure and supposedly the ability to reside in the UFP fraction. The average carbon oxidation state was not corrected for oxygen atoms that are attached to non-carbon atoms. Thus, compounds with oxygen containing functional groups (e.g. oxygen atoms

of the phosphate group in tricresyl phosphate) are shifted in the diagram towards seemingly higher $OS_C$-values, but in fact the carbon atoms exhibit a lower average oxidation state. Also in this illustration, we find no evidence for oxidation products of airport-related lubrication oil emissions. These results are consistent with the objective to design lubrication oils with a stability against oxidation. For this purpose antioxidants as N-phenyl-1-naphthylamine ($C_{16}H_{13}N$) and alkylated diphenyl amine ($C_{28}H_{43}N$) are added, which serve as radical scavengers (Wu et al., 2013). Even though tricresyl phosphate serves as an anti-

wear agent in lubrication oils, it is also reported that this additive is able to enhance the oxidation inhibition ability of antioxidant mixtures (Duangkaewmanee and Petsom, 2011).

We illustrate the Kendrick mass defect (CH2-base) vs. MW, in order to identify homologous series of hydrocarbons (Figure 3D). The Kendrick mass defect (KMD) is calculated by subtracting the Kendrick mass of the nominal mass (Equation 1). The Kendrick mass is defined as the IUPAC mass standardized on $^{12}CH_2$ with an exact mass of 14 Da (Kendrick, 1963).

$$KMD = \text{nominal mass} - (\text{IUPAC mass} * \frac{14.00000}{14.01565}) \qquad (1)$$

Thereby, compounds of a homologous series differing in the number of CH$_2$-groups align on a horizontal line. We observe the homologous series of pentaerythritol- and trimethylolpropane esters and their fragments as horizontal lines in Figure 3D.

### 3.2 Composition pattern of synthetic esters

We analysed the molecular pattern of the synthetic esters by comparison between the UFP samples and the five different jet

engine oils. Figure 4 shows the contribution of each individual ester molecule to the sum of all esters. The boxes illustrate the spread of the ester fraction in all 22 measured ambient UFP samples, and the coloured symbols indicate the composition of the oil standards. We observe that the ester composition of four out of the five jet engine lubrication oils matches the ester composition in the UFP samples well, which confirms that these esters can be attributed to the specific source of jet engine oils. Furthermore, the Mobil Jet Oil II follows most closely the ambient pattern of the median level of the pentaerythritol esters

(eight oxygen atoms), which is in line with the highest market share of this oil. The Eastman Turbo Oil 2380 consists of trimethylolpropane esters (six oxygen atoms) as the base stock, which are not a constituent of the other four jet engine oils. We observe a small but significant fraction of the trimethylolpropane esters in the ambient UFP, indicated by the consistently higher values of the boxes (and the median) compared to the pentaerythritol-ester-based oils. This is presumably a result of a minor utilization of trimethylolpropane ester containing jet engine lubrication oils in air traffic. Chromatograms of the various

jet engine lubrication oils are shown in Figure S5 and the individual molecular fingerprints as Figures S6-10, showing the different base stock materials.

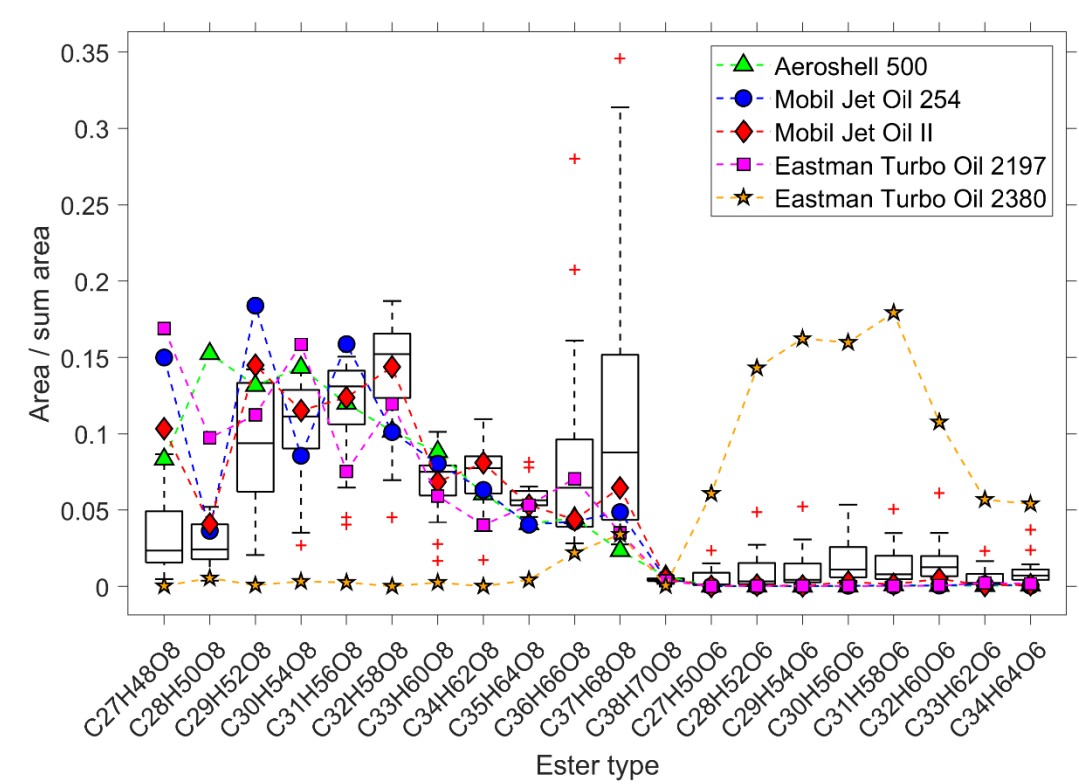

**Figure 4. Comparison of the pentaerythritol- and the trimethylolpropane ester ratios of five different jet engine oils (coloured symbols) with the observed spread in 22 ambient UFP samples (box plots). The y-axis is the ratio of the peak area of each individual ester to the sum area of all esters. The horizontal line within the box indicates the median, and the bottom and top edges of the box indicate the interquartile range. The whiskers show the spread to the most extreme values, and outliers (outside +/− 2.7σ) are shown as red "+" symbols.**

### 3.3 MS/MS fragmentation experiments

We used fragmentation experiments of the homologous series of pentaerythritol esters ($C_{27-38}H_{48-70}O_8$), N-phenyl-1-naphthylamine ($C_{16}H_{13}N$), alkylated diphenyl amine (Bis(4-(1,1,3,3-tetramethylbutyl)phenyl)amine; $C_{28}H_{43}N$), tricresyl phosphate ($C_{21}H_{21}O_4P$) and the homologous series of trimethylolpropane esters ($C_{27-34}H_{50-64}O_6$) in order to verify the identity of the molecules in the ambient UFP samples. These compounds have been described as molecular markers for jet engine lubrication oil emissions, and are not present in automotive lubrication oils (Fushimi et al., 2019). Figure 5 shows the comparison between the fragmentation patterns of tricresyl phosphate [A], the alkylated diphenyl amine [B] and one of the pentaerythritol esters [C] in the UFP samples and the Mobil Jet Oil II by ExxonMobil, respectively.

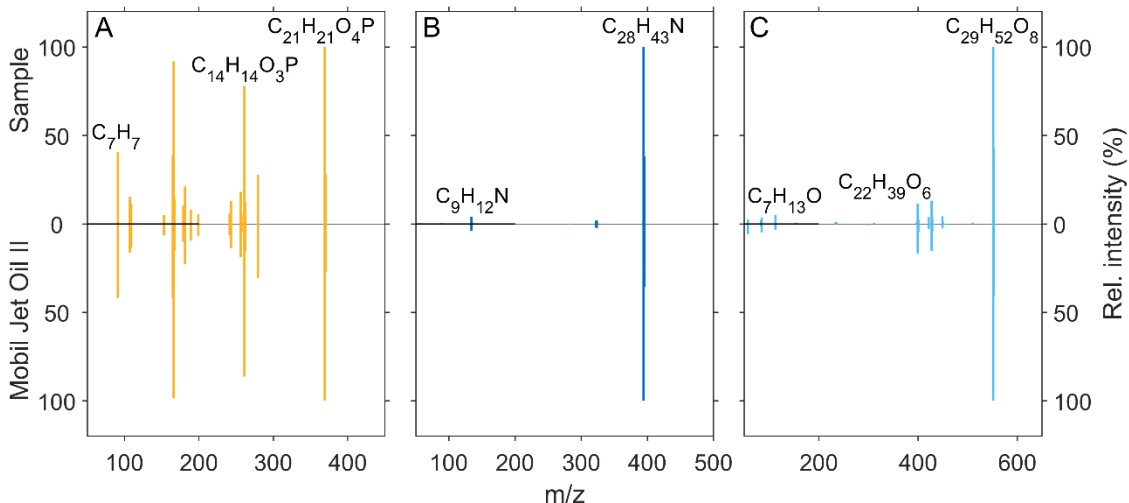

**Figure 5. Comparison of the MS/MS fragmentation patterns of tricresyl phosphate ($C_{21}H_{21}O_4P$), alkylated diphenyl amine ($C_{28}H_{43}N$) and one pentaerythritol ester ($C_{29}H_{52}O_8$) measured in an ambient UFP sample (0.032-0.056 µm) (upward spectra) and the Mobil Jet Oil II by ExxonMobil (downward spectra).**

The fragment at *m/z* 261.0674 ($C_{14}H_{14}O_3P$) is a fragment ion of tricresyl phosphate with only two aromatic moieties, whereas the fragment at *m/z* 91.0543 ($C_7H_7$) is the corresponding aromatic structure without the oxygen atom related to the phosphate group. Fragmentation of the alkylated diphenyl amine leads only to a slight formation of fragment ions, e.g. *m/z* 134.0964 ($C_9H_{12}N$), which is one of the two aromatic rings connected to the nitrogen atom and parts of one alkyl side chain. Hence it seems that the alkylated diphenyl amine additive features a higher stability compared to the other oil constituents with the used

fragmentation settings. The pentaerythritol ester ($C_{29}H_{52}O_8$), exemplary for the ester base stock material, shows a characteristic fragmentation pattern. The fragment ion at *m/z* 399.2737 ($C_{22}H_{39}O_6$) is an ester molecule after loss of one side chain ($C_7H_{13}O_2$) and the fragment ion at *m/z* 113.0960 ($C_7H_{13}O$) is an aliphatic side chain after ester bond cleavage. We were not able to obtain a clear fragmentation spectrum of N-phenyl-1-naphthylamine in the UFP samples as the detected signal intensity was not sufficient. This could be due to a minor usage of the additive in jet engine lubrication oils and a faster atmospheric degradation

rate, as it is utilized as a radical scavenger (Wu et al., 2013). Furthermore, the screening of the jet oils showed a matching MS/MS pattern of tricresyl phosphate and N-phenyl-1-naphthylamine with the mzcloud database.

**3.4 TCP isomer characterization**

Recent studies have reported that tricresyl phosphate exposure provokes detrimental health effects, as it exhibits a neurotoxic potential functioning as an acetylcholinesterase-inhibitor, and also by acting as an endocrine disruptor (Chang et al., 2020; Ji

et al., 2020). Different TCP isomers vary regarding their potential health effects for which reason it is important to gain information about the isomer composition of tricresyl phosphate in UFP. Therefore, we performed a targeted screening on tricresyl phosphate isomers using selected ion monitoring (SIM). The tri-*ortho* isomer shows a peak at 7.82 min, whereas the TCP peak of the UFP sample elutes after 8.06 min, likely an overlapping peak of a mix of different *meta-* and *para*-isomers.

The tri-*ortho*-isomer of tricresyl phosphate was not detected in the airport samples. This is in accordance with the reduction of

the *ortho*-isomer fraction of tricresyl phosphate used as an additive for lubrication oils (Winder and Balouet, 2002; Nola et al., 2008). Other studies have also reported that they were not able to identify the tri-*ortho* isomer of TCP in various sample types (Solbu et al., 2010; Solbu et al., 2011). Although no tri-*ortho* isomer of TCP was detected, it is still to consider that isomers with only one *ortho*-methyl group feature possibly a higher toxicity than ~~the~~isomers having methyl groups only in *meta*- and *para*-~~isomers~~position (Hanhela et al., 2005).

We detect further organophosphate compounds like triphenyl phosphate (TPP) in the UFP samples, which likely originate from hydraulic oils. TPP is a high production volume chemical used as a flame retardant, and likely has further sources than airport operations. However, Solbu et al. (2010) have reported that ground maintenance of aircrafts is also a possible source of organophosphates in the environment.

### 3.5 TMP-P formation from thermal decomposition of TMPE oils

Beside the common use of pentaerythritol esters as the base stock of lubrication oils, also trimethylolpropane esters (TMPE) are in use. It is known that a combination of TMPE with only 2% of a phosphate additive can result in the formation of the ~~neurotoxine~~neurotoxin trimethylolpropane phosphate (TMP-P) (Callahan et al., 1989; Centers, 1992). The combination of pentaerythritol esters with phosphate additives does not lead to the formation of TMP-P (Wyman et al., 1987). Studies have shown that the formation of TMP-P occurs possibly with all phosphates, regardless of the structure, at temperatures between

250-750 °C. The reaction is probably not highly temperature-dependent due to the low activation energy (Wright, 1996). At the upper temperature limit, the formed TMP-P starts to decompose, implying that the actual yield of TMP-P is depending on the balance between formation and degradation reactions (Callahan et al., 1989). TMP-P affects the nervous system by binding to γ-aminobutyric acid (GABA) receptors, especially at the picrotoxinin site and therefore affects neurotransmission processes (Bowery et al., 1976; Bowery et al., 1977; Mattsson, 1980; Simmonds, 1982; Ticku and Ramanjaneyulu, 1984).

In contrast to other studies (van Netten and Leung, 2000; Solbu et al., 2011), we detected TMP-P in the UFP samples, which could be the result of the continuing use of TMPE-based lubrication oils. TMP-P was identified via fragmentation leading to a characteristic phosphate fragment. The analysed Eastman Turbo Oil 2380 shows a different pattern regarding the jet engine oil marker compounds compared to the other tested oils (Figure 4). This oil is not based on a pentaerythritol, but on a trimethylolpropane ester formulation. Callahan et al. (1989) identified the Exxon 2380 oil as a source of TMP-P formation,

which is possibly the forerunner oil of the here tested Eastman Turbo Oil 2380. In order to confirm the identified TMP-P, we simulated the thermal decomposition in a temperature experiment using the Eastman Turbo Oil 2380. The jet engine oil was heated to temperatures of 250-450 °C in 50 °C steps for 4 h. Analysing the oil after the temperature treatment, we were able to detect an emerging signal at *m/z* 179.0467 ([M+H]$^+$, TMP-P, $C_6H_{11}O_4P$) at temperatures starting from 400 °C. The appearing compound features the same retention time as detected in the UFP samples, which speaks for the unambiguous identification

of TMP-P. To our knowledge, this is the first detection of TMP-P in ambient UFP. With increasing temperature, we observe the fraction of trimethylolpropane esters decreasing, whereas the fraction of TMP-P rises. The relative fraction of tricresyl

phosphate also increases, as it is temperature stable in contrast to the esters. The fraction of the two amines ($C_{16}H_{13}N$, $C_{28}H_{43}N$) decreases between 250-350 °C and from 400 °C onwards both compounds are not detectable anymore (Figure 6).

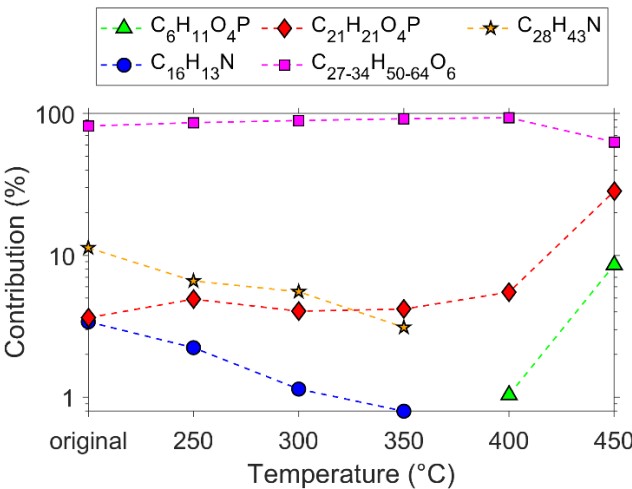

Figure 6. Development of the different constituents of the Eastman Turbo Oil 2380 during the temperature experiment. The contribution (%) is defined as the ratio of the area of one single compound relative to the total area of all jet oil constituents.

## 4. Conclusions

In this study, we demonstrate a novel analytical approach (non-target analysis) that reveals the chemical composition of UFP, down to the size range of 10-18 nm. We find that jet oil emissions contribute to the non-refractory fraction of particles <56 nm nearby Frankfurt airport, with all specified constituents detectable by HESI in the positive ionization mode. The analysis of the individual pattern of ester molecules and the comparison to jet oil standards revealed the presence of two different oil base stocks that emerge in the UFP at Frankfurt airport. Due to the identification of health-relevant organophosphorus compounds, future studies should perform toxicity testing and effect-directed chemical analysis in order to evaluate the health effects of UFP from large airports. In this context, it is essential to quantify the contribution of potentially harmful compounds that are present in airport-related UFP in future studies.

## Data availability

The data shown in this study is available by request to the corresponding author (vogel@iau.uni-frankfurt.de).

## Author contributions

FU wrote the paper, performed the field sampling, sample preparation and measurement, and did the bulk of the data analysis. DvP provided the Nano-MOUDI, advised on particle sampling, data interpretation and manuscript writing. ALV advised on data analysis, data interpretation and manuscript writing, edited the manuscript and directed the project administration.

**Competing interests**

The authors declare that they have no conflict of interest.

**Acknowledgements**

We thank Diana Rose, Florian Ditas and Stefan Jacobi of the Hessian Agency for Nature Conservation, Environment and Geology for providing access to the air quality monitoring station. We also thank Anett Dietze of the Leibniz Institute for Tropospheric Research (TROPOS) for the filter preparation.

**Financial support**

Funded by the Deutsche Forschungsgemeinschaft (DFG, German research foundation) project number 410009325.

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
