# Peer review of "Identification and Source Attribution of Organic Compounds in Ultrafine Particles near Frankfurt International Airport"

_Atmospheric Chemistry and Physics, 2020_

## Referee Comment (RC1) · Anonymous Referee #1 · 23 Nov 2020

This is a well written report of an important and timely study of the organic composition of particles emitted from aircraft operations near a major airport. The paper reports a careful study based on the UHPLC and HRMS analysis of collected PM samples, and interprets the consequences of the resulting compositional information. This is important since there have been many recent studies of UFP near airports, but the source apportionment and the particle composition has been lacking in most of those studies. This is an important advance in the field for this reason.

The reporting on organophosphate emissions adds to the understanding of these significant aircraft engine oil additives. In addition, I think that the authors have offered

some valuable new details of measuring organic species in aircraft exhaust and have also explored a new specific compound of potential interest and concern, TMP-P.

The paper is well written and the analysis careful and thorough. I have a few comments that, when addressed, I hope might provide improved clarity or otherwise improve the paper.

1) The analysis included accounting for sample blanks, where no sample was drawn. However no ambient background samples were acquired. I realize that the collection times are quite large. However, if the prevailing winds shifted for a long enough period, some background from a non-airport source would be useful to compare to the airport source. I presume that the long sample times preclude collecting sample in a trafficfree period with the prevailing wind direction, although that might even be a preferable ambient background target to another wind direction.

2) Line 38-39, forecasts are quoted for airline traffic growth. I think during the current pandemic, a sentence or two should mention that 2020 has seen a steep decline in air traffic and the recovery to pre-pandemic growth rates in highly uncertain in both time and value.

3) Lines 266-267 (and also abstract line 19), "... the sampled UFP are mainly composed of lubrication oils." Other studies quoted in this report have shown that lubrication oil is an important but not sole contributor to the organic component of aircraft PM. The paper goes on to say "However, other techniques might reveal the presence of additional compounds, which are not detected by the presented technique." Yet it is not clear what fraction of the organics can be attributed to aircraft engine oil. I think some further discussion should be included to either qualify this statement or to provide a quantitate estimate of the oil contribution, with whatever error bounds can be offered. Is UFP more than 50% composed of engine oil (I think not)? Can an estimate even be offered?

4) Line 357: I believe "neurotoxin" is misspelled as "neurotoxine".

---

## Referee Comment (RC2) · Anonymous Referee #2 · 3 Jan 2021

General comments. This manuscript presents an analysis of the chemical composition of the ultrafine particle fraction of aerosol samples collected downwind of a major airport. The authors used a non-targeted screening approach to identify $\sim$ 200 chemicals that were more than 5 times the signal observed in the blank. The majority of the compounds were attributed to jet engine lubricating oils based on molecular formula, MS/MS fragmentation, retention times in a UHPLC column, and a comparison to standards from jet engine oils. The data analysis is thorough and the paper is well written. However, there are places where additional information should be included and clarifications given. I recommend this for publication in ACP after the following specific comments are addressed.

[Figure]

Specific Comments 1. In the experimental it is stated that "Pure organic solvents provided the higher extraction efficiency than mixtures with water and similar ones to mixtures of acetonitrile and methanol." How was the extraction efficiency quantified and was it compared across different types of molecules, or only across the ones that are extracted well with the solvent that was chosen? What were the different pure organic solvents that were tested?

2. In the experimental it is stated that "A circular section with a diameter….according to the array of the nozzles of each impactor stage". Can you please clarify what this means? Were specific areas of the foil targeted?

3. In the discussion of the UHPLC/HRMS method it is noted that the two standards that were tested had a small linear response range "likely due to their adsorptive behavior on glass surfaces". If these are representative of the types of molecules found in this work, how likely is it that the other chemicals may have been influenced by this as well? I recognize that quantification was not attempted for the other compounds, but I would suggest adding a note to this effect given that qualitative comparisons of peak areas were carried out.

4. In the results and discussion it is noted that "the majority of these compounds does not distinguish from the blank". Does this mean that the majority of the compounds were also measured in the field blank? Were laboratory blanks also run and were these clean of the chemicals? Is this contamination occurring in the field, or is this coming during the sample handling/processing?

5. It is noted that the program provides a false assignment for the petaerythritol esters. On page 10 it is written that: "The native molecular fingerprint is displayed in Figure S4". What does this mean? Are these the results using the false assignments? If so, why is this being shown? If not, please clarify what this means and what is being plotted in Figure S4.

6. It is noted that the O/C ratios are below 0.6 and thus that the UFP do not become

oxidized during transport. Please provide the O/C range for the starting material to support this (from Figures S6-S10).

7. On page 15: "Although no tri-ortho isomer of TCP was detected, it is still to consider that isomers with only one ortho-methyl group feature possibly a higher toxicity than the meta- and para-isomers...". I am unsure what is being communicated here and suggest rephrasing.

Minor comments:

8. Please add a note that this is positive ion mode in the caption for Figure 2 (unless it includes both positive and negative ion mode, in which case please clarify that).

9. There is a darker purple in Figure 3 A and D that is not present in Figure 3 B and C. I think this is just a shading issue, but I recommend correcting it so that all the colors match the key.
* * *

---

## Author Comment (AC1) · 18 Jan 2021

**Reply to Anonymous Referee #1**

We thank referee #1 for the constructive comments. The original comments are in black font, our replies to the comments below appear in blue font, and changes in the manuscript are in red font.

**1**) The analysis included accounting for sample blanks, where no sample was drawn. However no ambient background samples were acquired. I realize that the collection times are quite large. However, if the prevailing winds shifted for a long enough period, some background from a non-airport source would be useful to compare to the airport source. I presume that the long sample times preclude collecting sample in a traffic-free period with the prevailing wind direction, although that might even be a preferable ambient background target to another wind direction.

Sampling representative ambient background samples is a difficult task. As the referee mentioned, the night-flight ban period is not long enough to accumulate sufficient mass for filter analysis. Another problem is that between 23:00 – 5:00 CET, although no flight movements are allowed, some aircrafts still arrive in this period due to delays and important medical cargo aircrafts that have an exceptional permission. This would lead to ambient background samples with contributions of aircraft emissions.
We also think that ambient background samples collected from another wind direction are not representative for the ambient background of Frankfurt airport when sampling during southerly wind direction, as further urban (north-east) and industrial (north-west) sources will contribute, that certainly do not contribute during airport sampling (south). The only way to go, to get a realistic picture of the background aerosol is to sample simultaneously north and south of the airport during southerhly wind direction. This approach however is not easily realizable, and could not be implemented during this study.

**2**) Line 38-39, forecasts are quoted for airline traffic growth. I think during the current pandemic, a sentence or two should mention that 2020 has seen a steep decline in air traffic and the recovery to pre-pandemic growth rates in highly uncertain in both time and value.

We agree- the global decline in air traffic 2020 due to the corona pandemic needs to be mentioned. We will also implement the decline in flight movements at Frankfurt airport in 2020, and update the given data about operations at Frankfurt airport from 2018 to the sampling campaign year 2019.

Line 39-41 (old version line 38-39): Due to the corona pandemic however, the European flight traffic in 2020 declined by 55% compared to 2019 (Eurocontrol, 2021). Current forecasts predict a full recovery of flight movements between 2024 and 2029, depending on the pandemic course (Eurocontrol, 2020).

Line 84-87: Frankfurt airport is one of the largest airports in Europe with more than 500,000 flight operations in 2019, shared over four runways. It is located in the Rhine-Main metropolitan area within a distance of around 12 km to the city centre of Frankfurt. In 2019 more than 70.5M passengers and 2.1M tons of cargo have been transported with a consumption of around $5.5 \times 10^6$ $m^3$ of kerosine (Fraport AG, 2020).

Line 87-88: The corona pandemic caused a decline in flight movements by 58.7% in 2020 compared to 2019. The transported cargo decreased by 8.5% and passenger numbers by 73.4% (Fraport AG, 2021).

**3)** Lines 266-267 (and also abstract line 19), "... the sampled UFP are mainly composed of lubrication oils." Other studies quoted in this report have shown that lubrication oil is an important but not sole contributor to the organic component of aircraft PM. The paper goes on to say "However, other techniques might reveal the presence of additional compounds, which are not detected by the presented technique." Yet it is not clear what fraction of the organics can be attributed to aircraft engine oil. I think some further discussion should be included to either qualify this statement or to provide a quantitate estimate of the oil contribution, with whatever error bounds can be offered. Is UFP more than 50% composed of engine oil (I think not)? Can an estimate even be offered?

Determination of the jet engine oil fraction in UFP is challenging, as it is not clear how efficient the Nano-MOUDI is sampling the UFP-fraction. Two possible types of losses / sampling artefacts can occur: (1) Wall-losses during sampling due to diffusion of the smallest particles, and (2) evaporation of semi-volatile compounds from the Nano-MOUDI stages due to the reduced pressure during sampling (100 mbar operating pressure on the smallest stage). Both processes can potentially reduce the fraction of jet engine oil in UFP in the given results, which would result in an underestimation of these compounds. Therefore, in the current stage we cannot give a quantitative estimate, since it requires a detailed compound-specific characterization of the nano-MOUDI sampling artefacts.
We like to rephrase our statement that based on our results it is suggested that UFP are mainly composed of jet engine lubrication oils. We soften the statement in line 266-267 (new file: 273-275), while we keep the statement in abstract line 19 unchanged, since it only describes our results in a qualitative manner.

Line 273-275 (old version l. 266-267): Hence, the results of the non-target analysis suggest that the sampled UFP are mainly composed of lubrication oils. However, other techniques might reveal the presence of additional compounds (e.g. metals, black carbon, inorganic compounds, etc.), which are not detected by the presented technique.

**4**) Line 357: I believe "neurotoxin" is misspelled as "neurotoxine".

Technical correction line 366 (old version: l. 357): neurotoxin

---

## Author Comment (AC2) · 18 Jan 2021

**Reply to Anonymous Referee #2**

We thank referee #2 for the constructive comments. The original comments are in black font, our replies to the comments appear below each comment in blue font, and changes in the manuscript are in red font.

Specific Comments **1**. In the experimental it is stated that "Pure organic solvents provided the higher extraction efficiency than mixtures with water and similar ones to mixtures of acetonitrile and methanol." How was the extraction efficiency quantified and was it compared across different types of molecules, or only across the ones that are extracted well with the solvent that was chosen? What were the different pure organic solvents that were tested?

In the revised paper we explain our extraction tests more in detail. The extraction tests were based on real ambient UFP filter samples extracted with different types of solvents. We minimized the consumption of our valuable filter samples, by only testing four different solvent combinations. The solvents were evaluated based on the non-target-software's identifications and corresponding signal intensities.

Line 131-134: Pure organic solvents provided the higher extraction efficiency than mixtures with water and similar efficiencies to mixtures of acetonitrile and methanol. The tests were based on UFP filter extractions with different solvents (100% methanol; 50% acetonitrile 50% methanol; 60% methanol 40% water; 60% acetonitrile 40% water), and evaluated based on the non-target-analysis-generated identifications and corresponding signal intensities.

**2**. In the experimental it is stated that "A circular section with a diameter: : :.according to the array of the nozzles of each impactor stage". Can you please clarify what this means? Were specific areas of the foil targeted?

For the extraction we cut out a section in the middle of the filters which is located below the array of nozzles of each impactor stage. By this approach we minimized the surface for extraction (e.g. the borders of the foil which were not exposed to particle deposition). We phrase this more clearly as follows:

Line 136-137: A circular section with a diameter of 2.5 cm was cut out of each foil sample located below the array of the nozzles of each impactor stage.

**3**. In the discussion of the UHPLC/HRMS method it is noted that the two standards that were tested had a small linear response range "likely due to their adsorptive behavior on glass surfaces". If these are

representative of the types of molecules found in this work, how likely is it that the other chemicals may have been influenced by this as well? I recognize that quantification was not attempted for the other compounds, but I would suggest adding a note to this effect given that qualitative comparisons of peak areas were carried out.

We agree- it is possible that the other compounds are influenced by adsorption to glass surfaces as well. As we do not quantify and just refer to detected compounds, this effect should be negligible regarding the given results. Since we compare the intensities of the different base stock esters, we will add a note that presumably all ester compounds show a similar adsorption tendency as the used ester standard.

Line 183-184: Presumably, all the detected pentaerythritol esters and trimethyolpropane esters show a similar adsorption tendency like the used ester standard.

**4**. In the results and discussion it is noted that "the majority of these compounds does not distinguish from the blank". Does this mean that the majority of the compounds were also measured in the field blank? Were laboratory blanks also run and were these clean of the chemicals? Is this contamination occurring in the field, or is this coming during the sample handling/processing?

In the non-target approach it is common to use field blanks to compare with collected samples, and it is not uncommon that the majority of detected compounds emerges from the procedure (especially in organic trace analysis). We did not analyse sample preparation laboratory blanks, as many different analysis steps would have to be taken into account. We did measure with each run a solvent blank, but this serves only to characterize the state of the instrument. The field blanks are the ones that were handled in the same way like the UFP samples, and therefore should represent different possible sources of contaminants (e.g. from sampling, transport, storage, filter preparation- and extraction procedure, and instrument). All compounds which do not show a signal-to-blank ratio larger than five were filtered out, and we observed no significant accumulation of these compounds during sampling, sample preparation and measurement. As all different contaminant processes are represented by the field blanks, we cannot identify from which step each contaminant emerges.

**5**. It is noted that the program provides a false assignment for the petaerythritol [sic] esters. On page 10 it is written that: "The native molecular fingerprint is displayed in Figure S4". What does this mean? Are these the results using the false assignments? If so, why is this being shown? If not, please clarify what this means and what is being plotted in Figure S4.

Yes the fingerprint in Figure S4 is the depiction of the results using the original non-target software results. With this plot in the supplementary information we wanted to highlight that software-generated results should not be trusted blindly without further evaluation, especially as non-target analysis of HR-MS data is a growing discipline that is used by many PhD students who are likely not familiar with potential cluster formation in (+)ESI-MS.

As we corrected the results of the software-generated analysis, we wanted to provide full transparency by also showing the native result.

**6**. It is noted that the O/C ratios are below 0.6 and thus that the UFP do not become oxidized during transport. Please provide the O/C range for the starting material to support this (from Figures S6-S10).

We will add a note regarding the O/C range of the purchased jet engine oils.

Line 282-283: The analysed jet engine lubrication oils show O/C ratios within the same range.

**7**. On page 15: "Although no tri-ortho isomer of TCP was detected, it is still to consider that isomers with only one ortho-methyl group feature possibly a higher toxicity than the meta- and para-isomers: : ". I am unsure what is being communicated here and suggest rephrasing.

As tricresyl phosphate (TCP) is composed of three aromatic moieties, six combinations regarding the position of the methyl groups are possible with at least one *ortho*-methyl group (o-m-m) (o-p-p) (o-m-p) (o-o-m) (o-o-o). We will formulate the sentence more clearly.

Line 356-358: Although no tri-*ortho* isomer of TCP was detected, it is still to consider that isomers with only one *ortho*-methyl group feature possibly a higher toxicity than isomers having methyl groups only in *meta*- and *para*-position (Hanhela et al., 2005).

Minor comments:

**8**. Please add a note that this is positive ion mode in the caption for Figure 2 (unless it includes both positive and negative ion mode, in which case please clarify that).

Thank you for the remark we will add the ionization mode in the caption of Figure 2.

Figure 1. The sample-to-blank ratios of the detected compounds, averaged over all samples belonging to a certain size fraction (0.010-0.018  $\mu$ m; 0.018-0.032  $\mu$ m; 0.032-0.056  $\mu$ m). Detection of the

compounds was accomplished in positive ionisation mode. The sample-to-blank ratio of 5 was specified for data filtering and is displayed as a dashed horizontal line.

**9**. There is a darker purple in Figure 3 A and D that is not present in Figure 3 B and C. I think this is just a shading issue, but I recommend correcting it so that all the colors match the key.

We have checked Figure 3 (A-D), every colored circle should be included in all plots. It is possible that a circle is visible in one plot (A) and not in another due to overlaying of isomers (appears naturally in B, C and D, while separated by chromatography in A). We did not use shading by creating the plots.